# MeDa-BERT: A medical Danish pretrained transformer model

**Jannik Skyttegaard Pedersen**[*]
The Maersk Mc-Kinney Moller Institute
University of Southern Denmark
jasp@mmmi.sdu.dk

**Martin Sundahl Laursen**[*]
The Maersk Mc-Kinney Moller Institute
University of Southern Denmark
msla@mmmi.sdu.dk

**Pernille Just Vinholt**
Department of Clinical Biochemistry
Odense University Hospital

**Thiusius Rajeeth Savarimuthu**
The Maersk Mc-Kinney Moller Institute
University of Southern Denmark

## Abstract

This paper introduces a medical Danish BERT-based language model (MeDa-BERT) and medical Danish word embeddings. The word embeddings and MeDa-BERT were pretrained on a new medical Danish corpus consisting of 133M tokens from medical Danish books and text from the internet. The models showed improved performance over general-domain models on medical Danish classification tasks. The medical word embeddings[1] and MeDa-BERT[2] are publicly available.

## 1 Introductions

Large language models (LLM) are powerful representation learners and have become the backbone structure of many modern natural language processing (NLP) systems. To learn text representations, LLM are first pretrained on a large-scale text corpus using self-supervised learning, e.g., masked language modelling. After pretraining, LLM are fine-tuned on specific downstream tasks where they have achieved state-of-the-art results on NLP benchmarks such as GLUE (Wang et al., 2018).

However, directly applying these general pretrained models to specialized domains such as the medical have led to unsatisfactory results (Peng et al., 2019). As a solution to this, a second round of in-domain pretraining (domain-adaptive pretraining) has shown to improve the performance of LLMs that were first trained on a general domain corpus (Gururangan et al., 2020). Domain-adaptive pretraining adjusts the weights of the

LLM to better capture the terminology, style, and nuances that are relevant to the target domain.

Resource-rich languages such as English have large domain-specific corpuses available that have been used to develop e.g., biomedical (Lee et al., 2020), clinical (Alsentzer et al., 2019), scientific (Beltagy et al., 2019), and financial (Peng et al., 2021) LLMs that perform better than models trained on general corpuses. These models could potentially be used to improve human decision making, save time, and reduce costs, e.g., by extracting information from scientific articles, identifying potential drug interactions, and helping with NLP tasks such as text classification, named entity recognition, and question answering for each of their specialized domains.

For the Danish language, only LLMs trained on a general domain have been made publicly available[3]. This paper presents a medical Danish BERT model (MeDa-BERT)—a LLM trained on a new medical Danish text corpus. We also used the medical corpus to train medical word embeddings as they still have value in the clinical domain (Laursen et al., 2023). To evaluate the medical word embeddings and MeDa-BERT, we used existing medical Danish classification datasets. We found that an LSTM model using the medical word embeddings outperformed a similar model using general-domain word embeddings, and that MeDa-BERT performed slightly better than a general-domain BERT model.

## 2 Method

This section first describes how the medical corpus was collected and used to pretrain the medical Danish word embeddings and MeDa-BERT. Next, the datasets used to compare model performances and the fine-tuning procedure is described.

---

[*]Equal contribution

[1]https://huggingface.co/jannikskytt/MeDa-WE

[2]https://huggingface.co/jannikskytt/MeDa-Bert

---

[3]Pedersen et al. (2022b) developed a clinical transformer model but it is not publicly available

| Corpus | Type | Date retrieved | Tokens |
|---|---|---|---|
| Clinical guidelines | Guidelines | October - November 2022 | 80,567,576 |
| Medicin.dk | Information portal | June 2021 | 28,878,335 |
| FADL | Books | January 2022 | 12,531,373 |
| Sundhed.dk | Information portal | May 2022 | 6,767,409 |
| Netdoktor.dk | Information portal | October 2022 | 3,227,051 |
| Wikipedia | Encyclopedia | October 2022 | 1,992,796 |
| **Total** | | | **133,964,540** |

Table 1: Number of tokens and date retrieved for each data source

## 2.1 Danish medical corpus

We collected data from the internet and from medical books. The owners of the data resources approved that we used their data in this study. We describe the data collection for each text contributor below. An overview of the text corpuses and their size can be seen in Table 1.

### 2.1.1 Clinical guidelines

We collected text from the document management systems of the five Danish regions. The documents contain guidelines and instructions for diagnostics and treatment of patients and all workflows that support this. The document systems also include non-medical documents from e.g. purchasing, logistics, and service departments which were removed. All departments that were excluded and the number of tokens retrieved from each region can be seen in Appendix A.

### 2.1.2 Medical information portals

We collected text from webpages that provide information to medical doctors and patients. The text was collected from Medicin.dk, Netdoktor.dk, and Sundhed.dk. The resources provide information about diseases, symptoms, and medical treatments. Moreover, the resources contain information specifically for health care professionals, e.g., medication guidelines and information about best practices in the field. Text not related to the medical domain and text written by non-professionals were removed from the corpus. A description of this process can be seen in appendix A.

### 2.1.3 Books

This part of the corpus consisted of 107 medical books from publisher FADLs Forlag that publishes books for medicine and nursing school.

### 2.1.4 Wikipedia

We used PetScan[4] to search for medical Wikipedia documents within predefined categories and its subcategories. We used a maximum depth of 5 for searching for subcategories. The following categories were used: anatomi, physiology, diseases, medication, epidemiology, diagnostics, medical procedures, medical specialities, medical physics, and medical equipment. We excluded documents with the categories: persons and companies. This process resulted in 5,391 documents. Next, we manually removed non-medical articles from that list which resulted in 5,266 documents.

## 2.2 Preprocessing of data

For all text corpusses, we defined a sample as one paragraph, i.e., a continuous stream of text without line breaks. We inserted spaces between alphanumeric and non-alphanumeric characters. Samples were further preprocessed to fit the pretraining procedure for either word embeddings or the transformer model, as detailed below.

### 2.2.1 Danish medical transformer model

MeDa-BERT was initialized with weights from a pretrained Danish BERT model[5] trained on 10.7 GB Danish text from Common Crawl (9.5 GB), Danish Wikipedia (221 MB), debate forums (168 MB), and Danish OpenSubtitles (881 MB).

For domain-adaptive pretraining, samples from the collected medical corpus were appended a [CLS] and [SEP] token in the start and end of each sample, respectively. Samples were concatenated to fit the maximum sequence length of 512 tokens and document boundaries were indicated by adding an extra [SEP] token in between samples. After this process, we removed duplicates corresponding to 0.2% of the total corpus. The model was trained using Adam (Kingma and Ba, 2015) with a weight decay of 0.01 as described in (Loshchilov and Hutter). Using gradient accumulation, the model was trained with a batch size of 4,032, a learning rate of 1e-4, and a linear learning rate decay warmed up over 1 epoch. The model was pretrained for a total of 48 epochs and evaluated after 16, 32, and 48 epochs. We used 5% of the samples as a validation set to evaluate the model during pretraining and trained the model on the remaining data using dynamic

---

[4] https://petscan.wmflabs.org/
[5] https://github.com/certainlyio/nordic_bert

| Dataset | Label | Train | Validation | Test |
|---|---|---|---|---|
| Bleeding | Positive | 10,331 | 1,300 | 1,300 |
| | Negative | 10,331 | 1,300 | 1,300 |
| Bleeding site | Airways | 1,000 | 125 | 125 |
| | Cerebral | 1,000 | 125 | 125 |
| | Ear-nose-throat | 1,000 | 125 | 125 |
| | Eyes | 1,000 | 125 | 125 |
| | Gastrointestinal | 1,000 | 125 | 125 |
| | Gynecological | 1,000 | 125 | 125 |
| | Internal | 1,000 | 125 | 125 |
| | Skin | 1,000 | 125 | 125 |
| | Urogenital | 1,000 | 125 | 125 |
| | Unknown | 1,000 | 125 | 125 |
| VTE | Positive | 9,064 | 1,100 | 1,100 |
| | Negative | 9,064 | 1,100 | 1,100 |
| VTE site | Airways | 1,600 | 200 | 200 |
| | Lungs | 1,600 | 200 | 200 |
| | Unknown | 1,600 | 200 | 200 |

Table 2: Dataset distributions

masked language modeling. The model was optimized using four Tesla v100 GPUs using the Huggingface (Wolf et al., 2020) library. All model parameters and pretraining losses are shown in Appendix B.

## 2.3 Danish medical word embeddings

We trained 300-dimensional FastText (Bojanowski et al., 2017) word embeddings. The embeddings were trained for 10 epochs using a window size of 5 and 10 negative samples. The hyperparameters were chosen to be able to compare the produced embeddings with the Danish FastText word embeddings from Grave et al. (2018) that were trained on a general domain.

## 2.4 Datasets

We compared performances between models using four medical datasets: bleeding classification, bleeding site classification, venous thromboembolism (VTE) classification, and VTE site classification. All samples were annotated with a consensus label from three medical doctors. The dataset distributions can be seen in Table 2 and examples of samples can be seen in Appendix C.

### 2.4.1 Bleeding classification

The bleeding dataset (Pedersen et al., 2021) is a binary classification problem with 25,862 samples. The dataset was constructed from 900 Danish electronic health records (EHR) from Odense University Hospital. The samples had an average token length of 13.3.

### 2.4.2 Bleeding site classification

The bleeding site dataset (Pedersen et al., 2022b) is a 10-class classification problem with 11,250 unique bleeding-positive samples annotated for the bleeding site. The bleeding site labels were: airways, cerebral, ear-nose-throat, eyes, gastrointestinal, gynecological, internal, skin, urogenital, and unknown. The dataset was constructed from 149,523 Danish EHR notes from Odense University Hospital. The samples had an average token length of 14.4.

### 2.4.3 VTE classification

The VTE dataset (Pedersen et al., 2022a) is a binary classification problem with 22,528 samples. The dataset was constructed from 94,520 Danish EHR notes from Odense University hospital. The samples had an average token length of 13.8.

### 2.4.4 VTE site classification

The VTE site dataset (Pedersen et al., 2022a) is a 3-class classification problem with 6,000 VTE-positive samples annotated for the VTE site. The VTE site labels were: airways, lungs, and unknown. The dataset was constructed from 94,520 Danish EHR notes from Odense University Hospital. The samples had an average token length of 14.5.

## 2.5 Fine-tuning
### 2.5.1 MeDa-BERT and BERT

We used the [CLS] token followed by a classification layer to classify samples of the datasets. We searched for the best models five times using Adam with learning rates [5e-5, 3e-5, 1e-5], i.e., we fine-tuned each model 15 times. The models were trained for a maximum of 10 epochs.

### 2.5.2 LSTM

We used the medical word embeddings as input to a bidirectional LSTM layer with a hidden layer size of 512. The last hidden state of the LSTM was followed by a dropout layer with probability 0.2, a dense layer of size 256, a ReLU activation function, a dropout layer of probability 0.2, and a dense classification layer. This model is referred to as LSTM+MeDa-WE.

The performance of the model is compared with another LSTM model (LSTM+General-WE) with the same parameters but using FastText embeddings trained on the general domain as input (Grave et al., 2018). We searched for the best

| | Bleeding | Bleeding site | VTE | VTE site |
|---|---|---|---|---|
| LSTM+General-WE | $83.8_7$ | $69.3_8$ | $88.5_2$ | $86.4_7$ |
| LSTM+MeDa-WE | $\mathbf{91.4_3}$ | $\mathbf{84.9_{1.1}}$ | $\mathbf{94.1_3}$ | $\mathbf{93.4_4}$ |
| BERT | $94.3_6$ | $86.7_8$ | $96.7_3$ | $94.7_3$ |
| MeDa-BERT_16 | $94.7_3$ | $88.4_6$ | $\mathbf{97.1_4}$ | $95.5_2$ |
| MeDa-BERT_32 | $95.1_5$ | $88.7_6$ | $96.9_3$ | $95.7_3$ |
| MeDa-BERT_48 | $\mathbf{95.3_4}$ | $\mathbf{89.1_2}$ | $97.0_5$ | $\mathbf{95.8_3}$ |

Table 3: Mean accuracy and standard deviation (subscript) for each model on four medical classification tasks. Best results for the LSTM and BERT-based models highlighted in bold. MeDa-BERT_16 denotes the MeDa-BERT model pretrained for 16 epochs.

models five times using Adam with learning rates [5e-5, 3e-5, 1e-5], i.e., we fine-tuned each model 15 times.

For all models we report the mean test set accuracy and standard deviation for the five best performing models on the validation dataset.

## 3 Results

Table 3 shows the results of each model on the four classification datasets.

### 3.1 Word embedding comparison

Using the medical word embeddings as input to an LSTM model resulted in large improvements compared to using general word embeddings. On average, LSTM+MeDa-WE outperformed the LSTM+General-WE model by 8.9 percentage points (PP). The largest improvement was seen on the 10-class bleeding site classification with an improvement of 15.6 PP.

### 3.2 Language model comparison

Comparing BERT and MeDa-BERT, the performance improvements were smaller. MeDa-BERT performed better on three of the datasets with an average improvement of 1.2 PP. The largest improvement was on the 10-class bleeding site classification with an improvement of 2.4 PP.

## 4 Discussion and limitations

This paper presented a new Danish medical corpus that was used to train NLP models. The corpus included medical books and text scraped from medical websites that provide information for both citizens and healthcare professionals. We applied different techniques to filter out non-medical data, e.g., by removing documents from non-medical

departments or text written by non-healthcare professionals. While these steps did remove a large part of non-medical text, some non-medical text might still be present in the corpus. However, the results showed that models pretrained on the medical corpus performed better than general-domain models, especially for multiclass classification problems.

For the Danish language, few medical evaluation datasets are available and therefore the models were only evaluated on classification tasks. Moreover, the evaluation datasets were constructed from EHR text which has its own nuances compared to the text of the medical pretraining corpus, e.g., EHR text contains many spelling mistakes whereas the medical corpus contains few grammatical errors. These factors might limit the generalizability of the results. Future work should evaluate the models on other tasks, e.g., named-entity recognition and question answering which will provide a better understanding of the models' capabilities.

We found continuous small performance improvements by pretraining MeDa-BERT for more epochs. The model might improve with further pretraining but because of limited computational resources and the small rate of improvement, we did not explore this further. The model would also benefit from more medical pretraining data. Although this paper presented a large part of the available medical Danish text, more data could be collected, e.g., from other medical book publishers and websites.

The medical datasets used to evaluate the models are not publicly available because of privacy concerns. For future work, we will strive to publish parts of the medical corpus which requires permission from the text owners. We advise interested researchers to contact us for sharing possibilities.

## 5 Conclusion

This paper presented a Danish medical corpus consisting of 133M tokens. The corpus was used to pretrain medical word embeddings and language models. The models trained on the medical corpus performed better than similar models trained on a general domain.

## Acknowledgement

We would like to thank the medical corpus contributors who gave acceptance of using their text in this project. Listed in alphabetical order: FADL's forlag, Medicin.dk, Netdoktor.dk, Sundhed.dk, The Capital Region of Denmark, The Central Region of Denmark, The Region of Northern Denmark, The Region of Southern Denmark, The Region of Zealand.

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

# Appendices

## A  Preprocessing of text corpuses

### A.1  Medical information portals

**Netdoktor.dk** provides information about diseases, symptoms, medication, and treatment. Netdoktor.dk contains sections that are not related to the medical domain and discussion forums where users can communicate. Therefore, we removed documents having links containing the following strings: debat, kultur, testdigselv, behandlerguiden, nyhedsbrev, nyheder, privacypolicy, kontaktnetdoktor, cookieinformation, disclaimer, sponsorindhold and discussions. Moreover, citizens can ask medical questions [6] that are answered by health care professionals. We only included the answers to these questions.

**Medicin.dk** has three sub-pages: `www.min.medicin.dk` that provides information to citizens, `www.pro.medicin.dk` that provides information to health care professionals, and `www.indlaegssedler.dk` that contains information about medicine. We included all documents from these webpages.

**Sundhed.dk** provides information for medical professionals [7] and citizens [8] about diseases, symptoms, medication and treatment. We included all documents from these webpages.

### A.2  Clinical guidelines

We collected clinical guidelines from the 5 regions of Denmark: The Capital Region of Denmark, The Region of Northern Denmark, The Region of

---
[6] `https://www.netdoktor.dk/brevkasser/`
[7] `https://www.sundhed.dk/sundhedsfaglig/`
[8] `https://www.sundhed.dk/borger/`

| Region | Categories removed (in Danish) | Date retrieved | Tokens |
|---|---|---|---|
| Capital Region | Den sociale virksomhed
Center for ejendomme
Center for HR
Center for Regional Udvikling
Region Hovedstadens Apotek
Steno Diabetes Center Copenhagena | October 2022 | 13,443,269 |
| Northern Region | Logistik afdeling
Teknisk Afdeling Himmerland
Teknik
Logistik
Service | October 2022 | 6,505,559 |
| Southern Region | Administration
Service
PsykInfo | September - November 2022 | 29,075,187 |
| Region Zealand | Administration
HR organisation og ledelse
Indkøb
IT
PortørCentral
Rengøring
Økonomi
Uddannelse | November 2022 | 6,387,083 |
| Central Region | | November 2022 | 25,156,478 |

Table 4: Categories removed and number of tokens from each of the Danish regions.

| Parameter | Value |
|---|---|
| **Architecture** | |
| Number of layers | 12 |
| Hidden size | 768 |
| FFN inner hidden size | 3072 |
| Attention heads | 12 |
| Attention head size | 64 |
| Dropout | 0.1 |
| Attention dropout | 0.1 |
| Max seq. length | 512 |
| **Optimization** | |
| Learning rate | 1e-4 |
| Optimizer | AdamW |
| Adam weight decay | 0.01 |
| Adam epsilon | 1e-6 |
| Adam beta1 | 0.90 |
| Adam beta2 | 0.98 |
| Learning rate decay | Linear |
| Batch size | 4032 |
| Warm up | 1 epoch |
| Epochs | 16, 32, 48 |
| Gradient clipping | 1.0 |

Table 5: Architecture and optimization parameters for pretraining MeDa-BERT

Southern Denmark, The Region of Zealand, and The Central Region of Denmark. For each region we removed non-medical documents, seen in Table 4.

## B  Model parameters and pretrainng loss

Table 5 shows the architecture and optimization parameters for pretraining MeDa-BERT. Table 6 shows the masked language modelling loss for MeDa-BERT during pretraining.

|  | Train loss | Validation loss |
|---|---|---|
| MeDa-BERT_16 | 2.122 | 2.019 |
| MeDa-BERT_32 | 1.874 | 1.792 |
| MeDa-BERT_48 | 1.766 | 1.673 |

Table 6: Masked language modelling loss for MeDa-BERT during pretraining. MeDa-BERT_16 denotes the model pretrained for 16 epochs.

## C   Dataset examples

Figure 7 shows a sample from each dataset translated from Danish to English.

| Dataset | Example | Label |
|---|---|---|
| Bleeding | "Girl hospitalized on 14.05.11 with bleeding tendency. 1½ years ago, noticed bleeding on both arms, under the armpits and on the inner thighs. Subsequently blood discharges on the mucous membrane of the cheeks and quite heavy menstrual bleeding, which is unusual for pt." | Positive |
| Bleeding site | "19-year-old man referred by on-call doctor due to sudden onset of macroscopic hematuria and left-sided flank pain." | Urogenital |
| VTE | "Pt has severe heart failure and hence dyspnoea and the feeling of air hunger, and, in addition, pt has pulmonary embolisms and COPD. Treatment with Fragmin has started." | Positive |
| VTE site | "Irregular contours on the left side in the transverse sinus and beginning part of the sigmoid sinus compatible with partial thrombosis." | Brain |

Table 7: Example of a sample from each dataset translated from Danish to English.