# OpenReview forum: "MeDa-BERT: A medical Danish pretrained transformer model"
_NoDaLiDa/2023/Conference — NoDaLiDa 2023_

### Official Review · Reviewer_B4ju · 2023-03-02
**The article describes the creation of categorizers for a set of diseases or conditions. The authors trained BERT models on Danish medical texts and report a high classification performance on their test sets.**

**Rating:** 8
**Confidence:** 4

**Review:**

The article describes the creation of categorizers for a set of diseases or conditions. The authors first collected a large corpus of medical texts in Danish from which they trained a BERT model: MeDa-BERT. They initialized the weights with those of another BERT model trained with 10 epochs (if I understood well. This is not clear from the text). They also trained FastText embeddings on this corpus.

The authors then used a four corpora of medical documents annotated with one disease or condition. Two corpora have binary classes, while two others have respectively 10 and 3 classes.

They authors evaluated two architectures with different parameters. The first one is an LSTM network with either nonspecialized embeddings or embeddings trained on the medical corpus. The second one is a transformer with different BERT configurations.

The authors found that the medical fine-tuning yields better results for LSTMs as well as for the BERT transformers, BERT being more accurate than LSTM. They also found that more training epochs improves the performance.

One point is unclear: The number of epochs, where the authors write 10 in Sect. 2.5.1. Does it apply to the transformer head only or to the Danish BERT pretraining? If this is for BERT, the comparison in Table 3 would be unfair at the medical BERTs have been trained with up to 48 epochs. In addition, MeDa-BERT has more epochs by construction as its starts from initialized weights. The authors should make this clear and, maybe, soften, or discuss, their claims of better performance.

Overall a good paper. I recommend to accept it.

**Paper Type:**

Short paper

---

### Official Review · Reviewer_svsv · 2023-03-04
**A short paper presenting a BERT model trained on Danish medical data and the corresponding word embeddings**

**Rating:** 7
**Confidence:** 4

**Review:**

A interesting read on a new BERT model trained on Danish, a somewhat underrepresented language in the field, and more specifically on medical data. The authors argue for the need of this in downstream tasks like NER and text classification for specialised domains where simple pre-trained BERT models often fail.

Pros:
1. Subsection 2.1 mentions data owner consent which is crucial when working with this kind of data, even if they are "publicly" available
2. Clear explanation of the data and the preprocess steps
3. Clear explanation on the pre-training of the model, and the embeddings good for replication

Cons:
1. The training of static word embeddings was not so justified. What is their value and use cases?
2. Table 3: Difference between LSTM using general embeddings and an LSTM using the paper's embeddings is more clear (at least for the bleeding site dataset), than the comparison of BERT and MeDA BERT which does not seem that significant
3. Lack of error analysis and concrete examples to make the reader understand the difference between the proposed word embeddings and BERT model and what is already available, especially seeing how the difference in performance is not that high. I understand the data are private, but pseudo examples or anonymised examples could have been used.

All in all, I think this is a good paper touching upon the issue of using pre-trained LMs in domain specific tasks and for languages like Danish which are not as well represented as English is.

**Paper Type:**

Short paper

---

### Official Review · Reviewer_ryvg · 2023-03-10
**A rather good paper, but with some notable shortcomings**

**Rating:** 6
**Confidence:** 4

**Review:**

This paper presents a substantial Danish medical corpus consisting of 133M tokens, as well as word embeddings and a BERT language model, both pre-trained on this corpus and evaluated on a specific text classification task, using existing Danish medical classification datasets.

Although the title of the paper focuses on language modeling, putting the pre-trained MeDa-BERT model as the central result of this research, I would say that the central result is the Danish medical corpus, which has been systematically created and evaluated.

As evaluation shows, the MeDa-BERT model performs slightly better than a general-domain Danish BERT, however, it is not compared to a recently developed and published Danish Clinical ELECTRA model (Pedersen et al., 2022), making impression that this is the first Danish transformer model for the medical domain (078: "For the Danish language, only LLMs trained on a general domain have been published.").

As for evaluating the medical word embeddings with an LSTM model (vs. a similar model with general word embeddings), medical word embeddings significantly outperform the general ones (for a medical classification task), but the LSTM architecture is not a competitor to a transformer model, as expected.

Overall, the paper can be very useful for those creating medical corpora for less resourced langauges, while the evaluation is limited to a rather specific dataset (even within the medical domain) of very short texts, and there is no comparison made to the above mentioned Danish medical transformer model. Thus, it also raises a question whether it is indeed the first large medical corpus for Danish (410: "This paper presented the first Danish medical corpus..").

**Paper Type:**

Short paper

---

### Decision · Program_Chairs · 2023-03-17

Accept